# A Neural Model for Word Repetition

**Daniel Dager[*], Robin Sobczyk[*], Emmanuel Chemla, Yair Lakretz**

École normale supérieure, PSL University, EHESS, LSCP, CNRS, Paris, France

[*]Contributed equally

## Abstract

**It takes several years for the developing brain of a baby to fully master word repetition — the task of hearing a word and repeating it aloud. Repeating a new word, such as from a new language, can be a challenging task also for adults. Additionally, brain damage, such as from a stroke, may lead to systematic speech errors with specific characteristics dependent on the location of the brain damage. Cognitive sciences suggest a model with various components for the different processing stages involved in word repetition. While some studies have begun to localize the corresponding regions in the brain, the neural mechanisms and *how* exactly the brain performs word repetition remain largely unknown. We propose to bridge the gap between the cognitive model of word repetition and neural mechanisms in the human brain by modeling the task using deep neural networks. Neural models are fully observable, allowing us to study the detailed mechanisms in their various substructures and make comparisons with human behavior and, ultimately, the brain. Here, we make first steps in this direction by: (1) training a large set of models to simulate the word repetition task; (2) creating a battery of tests to probe the models for known effects from behavioral studies in humans, and (3) simulating brain damage through ablation studies, where we systematically remove neurons from the model, and repeat the behavioral study to examine the resulting speech errors in the "patient" model. Our results show that neural models can mimic several effects known from human research, but might diverge in other aspects, highlighting both the potential and the challenges for future research aimed at developing human-like neural models.**

**Keywords:** word repetition; deep learning; speech errors; working memory

## Introduction

Across multiple cognitive domains, from perception to language and reasoning, *dual-route processing* appears as a key computational strategy of the human brain to address complex tasks (Marshall & Newcombe, 1973; Dell et al., 1997; Tversky & Kahneman, 1974; Mishkin et al., 1983; Hickok & Poeppel, 2007; Evans, 2008). Dual-route processing relies on the combination of a *memory-based* and a *rule-based* route, which are used to process incoming information and choose subsequent actions. The memory-based route draws on Long-Term Memory (LTM) to handle familiar information from past experiences. This type of processing is often fast, automatic, and unconscious, making it highly efficient. In contrast, the rule-based route relies on fresh computations and Working Memory (WM), which is slower but crucial for processing novel information. This idea of dual-route processing can be traced back to the work of William James, who distinguished between actions selected based on habit and those that involve effortful deliberation (James, 1890).

A dual-route processing account has been proposed for numerous tasks, including word repetition (e.g., Goldrick & Rapp, 2007; Nozari et al., 2010). Word repetition involves hearing a word – whether real or a pseudoword – and repeating it aloud. For healthy adult speakers, this is typically a simple task that rarely leads to errors, however, it could present challenges for various populations. For example, babies and toddlers, still developing their language skills, often find word repetition difficult as they are in the early stages of processing and producing speech. Similarly, children with developmental disorders may struggle with this task in specific ways. In adults, individuals who have experienced neurological events such as a stroke may face varying degrees of difficulty with auditory word repetition. Studying these challenges provides valuable insights into the intricate cognitive and neural processes involved in language. In fact, research on the deficits observed in patients has contributed to the development of an information-processing Cognitive Model for Word Repetition (Dotan & Friedmann, 2015).

The cognitive model for word repetition has two main routes (Figure 1A-Top): a *lexical route* and a *sublexical route*. Both routes begin by processing the acoustic input. The lexical route is used for familiar words, activating stored information from long-term memory (LTM), which can be used for their pronunciation. In contrast, the sublexical route handles new words that are not yet stored in LTM, relying on rules to convert a sequence of phonemes into their sequential production. The lexical route is typically fast and efficient, leveraging LTM, and the sublexical route is slower and constrained by WM limitations. However, the sublexical route is crucial for language acquisition in children as well as in adults (e.g., Susan E. Gathercole & Emslie, 1994).

Neuroimaging studies have identified neural pathways that may correspond to these routes. The ventral stream is involved in lexical processing, while the dorsal stream handles sound-to-motor mapping (Hickok & Poeppel, 2007; Rauschecker & Scott, 2009), and damage to this route can lead to impaired speech repetition (Fridriksson et al., 2010). While these studies offer evidence for where word repetition may occur in the brain, the underlying neural mechanisms involved in each processing stage remain largely unknown. Here, we move towards linking the cognitive model to neural mechanisms in the brain by modeling the task using neural networks that simulate dynamics which resemble those of

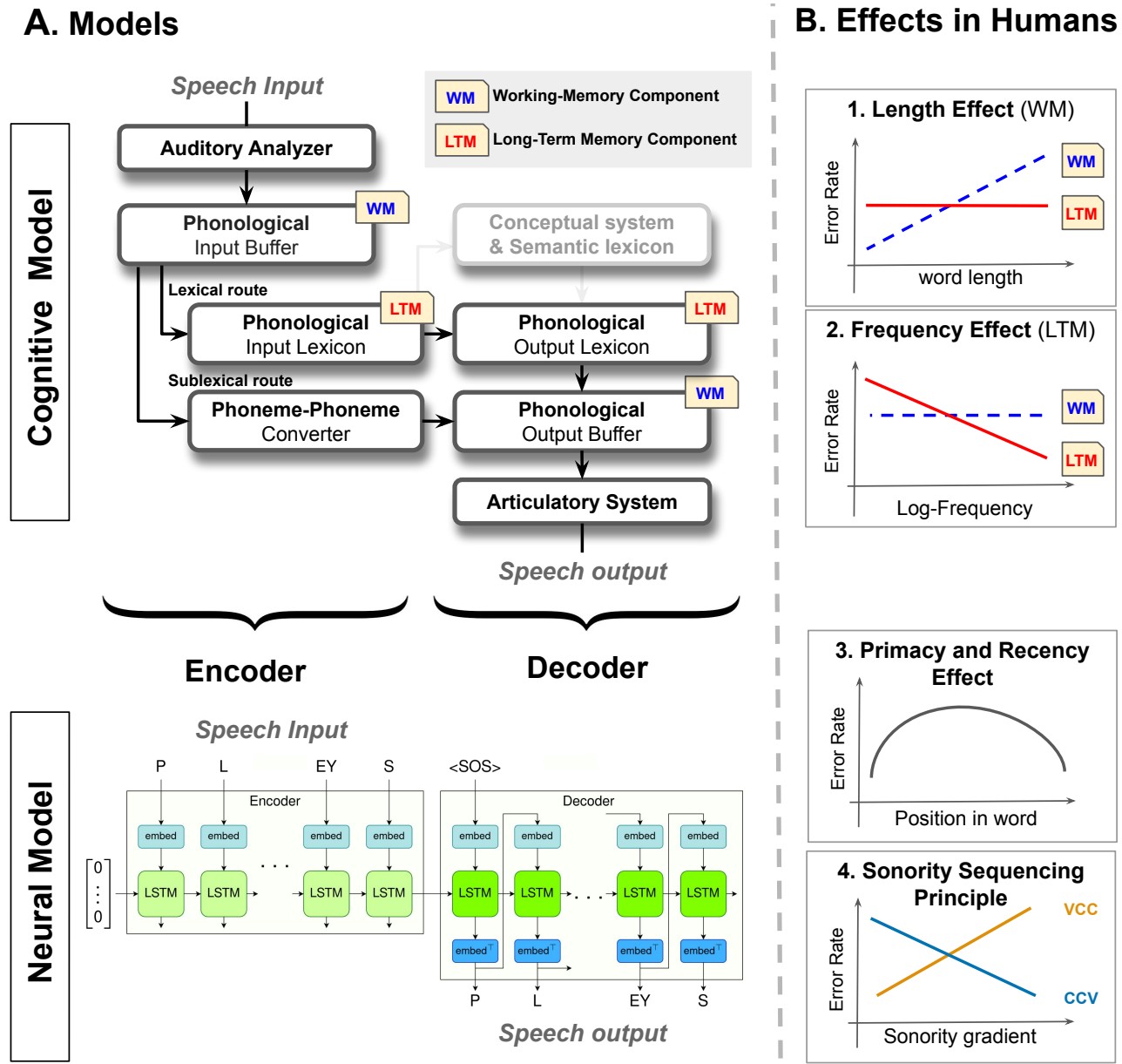

Figure 1: **Linking the Cognitive Model for Word Repetition and Brain Dynamics with Deep Neural Models.** (A) **Top**: A diagram of the cognitive model for word repetition, illustrating the various underlying processing stages. The so-called '**Buffers**' represent working memory (WM) components, which have capacity limitations and can only store information temporarily. The so-called '**Lexicons**' in the model represent long-term memory (LTM) components, which can store tens of thousands of words and allow for retrieval during processing. **Bottom**: An encoder-decoder architecture used to model word repetition with a neural network. (B) Known effects from human research: (1) **Length Effect:** A tendency to make more errors on longer words. This effect is observed in WM components but not in LTM components, due to the capacity limitations of WM. (2) **Frequency Effect:** The tendency to make fewer errors on words that are more frequent in language. In contrast to the length effect, this effect is observed in LTM component but not in WM components – frequent words are stored and retrieved more efficiently in LTM. (3) **Primacy and Recency Effects:** The tendency to make fewer errors on phonemes at the beginning (primacy) and at the end (recency) of a word. Phonemes at the beginning of a word are often more easily encoded and retrieved due to their prominence in speech perception, while phonemes at the end of a word benefit from more recent activation in working memory. (4) **Sonority-Sequencing Principle:** The principle states that sonority increases towards the nucleus of a syllable (typically the vowel) and decreases afterwards. In CCV structures (C: consonant, V: vowel), consonant clusters show a gradient where the sonority rises towards the vowel, while in VCC structures, sonority decreases towards the final consonant.

the brain. Unlike biological networks, these neural networks are fully observable, which offers an opportunity to study the mechanisms behind word repetition.

We used an Encoder-Decoder (aka, seq2seq; Sutskever, 2014) architecture with recurrent neural networks (RNNs), which captures the two main parts of the cognitive model (Figure 1A-Bottom). We trained a large set of models to perform the word-repetition task on the full English vocabulary, weighted by word frequency. We then studied the models behaviorally, and asked whether the errors of the models mimic known phenomena from human studies. We finally studied the models neurally through ablation studies to identify the role of specific subsystems: we studied the errors of our 'patient' models, asking whether they resemble speech-error patterns akin to those identified in human patients, and whether dual-route processing naturally emerges in an otherwise generic neural architecture.

The main contributions of our study are: (1) Encoder-Decoder neural models that perform the word-repetition task; (2) A suite of tests to study human-like processing in the models; (3) A framework to examine if dual-route processing emerges spontaneously in a generic neural network as it learns; (4) 'Patient' models that simulate speech errors in human patients. [1]

## Related Work and Background

### The Dual-Route Processing for Word Repetition

Evidence for the dissociation between two pathways in word repetition comes from neuropsychological studies, which identify two groups of patients with distinct error patterns. One group produces errors indicative of lexical processing, such as sensitivity to word frequency, while the other produces errors indicative of sublexical processing, such as sensitivity to syllabic structure or phoneme frequency (e.g., Goldrick & Rapp, 2007; Nozari et al., 2010). These findings were integrated into a cognitive model for word repetition (Figure 1A).

In this model, the lexical and sublexical routes share a common initial stage in the so-called Auditory Analyzer, where phoneme identities and positions are extracted from word acoustics. This information is transiently held in the *Phonological Input Buffer*. In general, so-called 'buffers' of the model are WM components[2], which store information for a relatively short time. Due to their limited capacity, they typically show length effects. Information from the Phonological Input Buffer then flows into the two main routes of the model, the lexical and the sublexical routes.

The lexical route involves accessing and retrieving entries from LTM, which are stored in two lexicons. The *Phonological Input Lexicon* stores auditory representations of entire words, and the *Phonological Output Lexicon* stores more abstract

representations that are shared with also other tasks, such as reading and naming (Marshall & Newcombe, 1973; Friedmann & Coltheart, 2018; Dotan & Friedmann, 2015). Evidence for selective impairments of each of these lexicons, and therefore to their separate existence, comes from neuropsychological studies, showing such double dissociation (e.g., Shallice, 1981; Caramazza & Hillis, 1990).

In contrast to the lexical route, the *sublexical route* directly maps input to output phonology, bypassing the lexical system, through a set of conversion rules. These rules control the mapping of short sequences of heard phonemes during word comprehension onto the corresponding sequences for word production. The sublexical route is used to process new words. Since new words lack lexical entries, they cannot be processed fully through the lexical system.

Both routes converge at the *Phonological Output Buffer*, which is the stage in language production where phonemes are held in working memory and assembled into words (Romani, 1992; Vallarb et al., 1997; Shallice et al., 2000). It serves two primary functions, first, as a phonological working memory that maintains phonological information until articulation, and second, it assembles phonemes into words and combines stems and affixes into complex words (Dotan & Friedmann, 2015; Haluts et al., 2020). This stage therefore has a key role across several word-processing tasks: naming, reading, and repetition of both words and pseudowords.

### Word-Processing Phenomenology

Research on both healthy individuals and patients has revealed several key insights into word repetition. Here, we focus on four established effects. To these, we add two more: one derived from typical WM characteristics and another from phonological theory. These six effects will guide the analyses of the neural models (Figure 1B):

1. **Lexicality Effect: Pseudowords (non-words that follow phonological rules but have no meaning) are more prone to errors than real words.** This effect is key for differentiating lexical vs. sublexical processing in the two routes, since pseudowords are necessarily processed through the latter route.

2. **Frequency Effect: Low-frequency words are more prone to errors than high-frequency words.** This effect is key for differentiating *lexicons* from *buffers* in the models, since lexicons, as LTM components, but not buffers, are predicted to show frequency effects.

3. **Length Effect: Longer words are more prone to errors than short words.** This effect is key for differentiating buffers from lexicons in the models, since buffers, as WM components, but not lexicons, have limited capacity (Baddeley et al., 1975).

4. **Morphological-Complexity Effect: Morphologically-simple words are more prone to errors than equi-length morphologically-complex words.** Morphemes (e.g., 'ing'

---

[1] All necessary materials, stimuli and scripts necessary for reproduction can be found at https://github.com/danieldager/swp-model

[2] While certain processes might align more with short-term memory (STM), working memory (WM) and STM are used synonymously for ease of exposition.

or 'able') were shown to be stored as basic units, like phonemes, function and number words (Dotan & Friedmann, 2015), from which the phonological output buffer composes words. The *effective* length of morphologically-complex words is therefore shorter than that of morphologically simple words of equal length, making them less prone to errors related to word length.[3]

5. **Primacy and Recency Effect: Phonemes in middle positions of words are more prone to error than early and late positions.** A well-established phenomenon in working memory is the serial position effect (Murdock Jr, 1962). In a sequence, items presented at the beginning are better retained due to their saliency, known as the *primacy effect*. Items presented at the end of the sequence are also more easily recalled due to their recency during retrieval, known as the *recency effect*. In contrast, items that appear in the middle of the list tend to be forgotten more often. This primacy and recency effects have been consistently demonstrated in tasks such as free recall and immediate serial recall (ISR), as well as in pseudoword repetition tasks (e.g., Hartley & Houghton, 1996; P. Gupta, 2005; P. Gupta et al., 2005; Page & Norris, 2009).

6. **Sonority-Gradient Effect: consonant clusters that violate the Sonority Sequencing Principle are more prone to errors than consonant clusters that obey it.** The Sonority Sequencing Principle (SSP; Selkirk, 1984; Clements, 1990) describes how syllables are structured based on the sonority, or loudness, of sounds. It suggests that the central part of a syllable, typically a vowel, is the peak of sonority, and the surrounding consonants should have progressively lower sonority as you move away from the vowel. For example, in the English one-syllable word "plant", the consonants "p" (low sonority) are followed by "l" (high sonority), and the vowel "a" forms the peak of sonority, with the consonants "n" (high sonority) and "t" (low sonority) completing the syllable. While many languages follow this pattern, some languages, allow for violations of this rule. English follows the SSP but also has exceptions, such as the /s/ + stop clusters (e.g., in 'sport'). Overall, we expect more repetition errors for phoneme sequences that violate the SSP.

### Computational Models for Word Repetition

Our approach draws on influential prior computational models of language processing and short-term memory (e.g., McClelland et al., 1989; Dell et al., 1997; Botvinick & Plaut, 2006; S. Gupta et al., 2020; Sajid et al., 2022). More recent work has attempted to model word repetition by incorporating knowledge of the neuroanatomy of the language system (Ueno et

al., 2011; Chang & Lambon Ralph, 2020). However, these models were trained on a relatively small vocabulary—only a few hundred words—far smaller than the lexicon of an average speaker. Additionally, all words were restricted to monosyllables. Here, we leverage advances in machine learning to train a deep neural model on the full lexicon, achieving perfect performance. This enables the use of richer probing datasets that are not limited to monosyllabic words and allows for the exploration of length and morphological effects.

## Experimental Setup

### Datasets

**The Training Dataset**  Comprises the 30K most frequent English words, based on the WordFreq python library (Speer, 2022). We excluded abbreviations and words that were not found in the CMU dictionary (CMU, 2014). Each word was included at least once in the dataset, after which words were sampled by frequency, with replacement, in order to generate $10^6$ total samples. The CMU dictionary provided us with the ARPAbet phonetic translation of each word, including vowel stress, which, for simplicity, we do not model in this work.

**The Word Feature Evaluation Dataset**  Given the known processing effects from humans (*Section - Word-Processing Phenomenology*), we created a factorial design with four main dimensions, which allows for the disambiguation of the effects of interest: lexicality effect, morphological-complexity effect, length effect, and frequency effect (see Table 1). The evaluation dataset has 100 words for each of the 12 conditions, summing to a total of 1200 words. The factorial design allows for splitting the dataset according to any one condition (e.g., 600 short words vs. 600 long words). Real words were selected from the training dataset. Pseudo-words were generated using an algorithm that leverages the trigram statistics of the training dataset (New et al., 2004). To enhance sublexicality, we included only pseudowords that were orthographically far from all real words in the training dataset. To quantify this, we computed the Levenshtein edit distance to all real words and normalized it by pseudoword length. We then included pseudowords whose minimal length-normalized edit distance to all real words was at least $0.25$. Meaning, four-letter pseudowords could share all but one phoneme with any real word, whereas eight-letter pseudowords needed to differ by at least two. Finally, the phonetic transcriptions of all pseudowords were generated using the G2P python library (Park & Kim, 2019). As with the training dataset, vowel stress was removed.

**The Sonority Evaluation Dataset**  To explore whether error rates correlate with the phonotactics of the language, we created a dataset with all the possible consonant-consonant-vowel (CCV) and vowel-consonant-consonant (VCC) combinations, excluding combinations where the same consonant was repeated and those that were in the training dataset. We then quantified the sonority gradient in the resulting syllables by computing the difference between the phoneme classes of

---

[3]While morphologically complex words might be less prone to length-related errors than equi-length monomorphemic words due to their smaller "effective size", they nonetheless exhibit specific error patterns, particularly morphological errors such as affix omissions, substitutions, or insertions, as described in Dotan & Friedmann (2015). These specific error types will not be explored in the current work.

| # | Lex. | Morph. | Length | Freq. | Example |
|---|------|--------|--------|-------|---------|
| 1 | Real | Simple | Short | High | Boot |
| 2 | Real | Simple | Short | Low | Clog |
| 3 | Real | Simple | Long | High | Prestige |
| 4 | Real | Simple | Long | Low | Gauntlet |
| 5 | Real | Complex | Short | High | Undo |
| 6 | Real | Complex | Short | Low | Anew |
| 7 | Real | Complex | Long | High | Restart |
| 8 | Real | Complex | Long | Low | Joviality |
| 9 | Pseudo | Simple | Short | N/A | Quab |
| 10 | Pseudo | Simple | Long | N/A | Curroxima |
| 11 | Pseudo | Complex | Short | N/A | Rebo |
| 12 | Pseudo | Complex | Long | N/A | Deoborer |

Table 1: **The Word Feature Evaluation Dataset.** We created a factorial design to probe the models, which has four main dimensions: (1) Lexicality, (2) Morphological Complexity, (3) Word Length and (4) Word Frequency. An example is given for each of the 12 conditions. The dataset contained 100 samples from each condition.

the adjacent consonants. That is, following the SSP, we first ordered phoneme classes based on their sonority: $glide(1) > liquid(2) > nasal(3) > fricative(4) > plosive(5)$. Then, for each consonant cluster, we computed the difference between the two classes as the difference between their rank in this order. For example, if the first consonant was plosive and the second one was nasal, then the sonority gradient was set to $5 - 3 = 2$, and it was set to $3 - 5 = -2$ if the first was nasal and the second was plosive. This means that CCV syllables with a positive sonority gradient follow the SSP and those with a negative gradient violate it; and vice versa for VCC syllables.

## Models

**Architecture**  We used a standard Encoder-Decoder architecture (Sutskever et al., 2014), with either simple (Elman) or Long-Short Term Memory (LSTM; Hochreiter & Schmidhuber, 1997) units, see Figure 1A-Bottom.[4]

**Encoder-Decoder**  The Encoder first passes the tokens through an embedding layer, and then through the recurrent layer (or layers, of RNN or LSTM units). The final hidden state of the Encoder was passed as the initial state of the recurrent layer of the Decoder. For simplicity, the Encoder and Decoder always had the same unit type, hidden size and number of layers. At each time step in the Decoder, the previous output was fed back to the recurrent network as the input for the next token prediction. The first input embedding of the Decoder was the Start-of-Sequence token. The weights for the input embedded layers in Encoder and Decoder were shared, and the output embedding layer of the Decoder used their trans-

---

[4]Our choice of recurrent architectures over more recent Transformer models is driven by their greater biological plausibility, as they more closely resemble the sequential processing of biological brains, while still offering robust performance for the tasks explored here. Notably, some recent advancements in neural network architectures, such as State Space Models (SSMs; Gu & Dao (2023)), are exploring a return to designs that incorporate forms of recurrent connections.

position. After being embedded, tokens are passed through a dropout layer.

**Training Procedure**  LSTM models were trained for 100 epochs, at which point our best models had perfectly learned the training data. RNN models required more epochs to converge, and were trained for 150 epochs, but ultimately failed to achieve zero error rate on the training data. We used the standard ADAM optimizer (Kingma & Ba, 2014), and a variation of Cross-Entropy Loss which allowed us to ignore the pad tokens we used to align the sequences for batching.

**Model Selection**  After a preliminary parameter search, we ran a finer grid search over models with a single layer (see Table 2 in Appendix). For model selection, we used the following criteria: (1) perfect accuracy on the CV training splits; (2) highest accuracy on the CV validation splits; (3) smallest model complexity, in terms of number of parameters.

## Analyses

**Measures**  We used two measures for model performance: (1) Error rate, the fraction of words that the model fails to perfectly repeat, and (2) Edit distance, the average of the Levenshtein distances (Levenshtein, 1965) between each predicted phoneme sequence and its corresponding ground truth.

**Model Evaluation**  After training and model selection, we computed the error rate and the edit distance of the selected models on the Word Feature Evaluation Dataset and on the Sonority Evaluation Dataset.

**Behavioral Study**  To determine which factors—lexicality, length, and morphological complexity—best predict model performance, we regressed model performance on all factors, including interaction terms. Since regression coefficients are sensitive to possible correlations among factors, we also conducted a Feature-Importance (FI) Analysis for the main effects, which is more robust to such correlations (Breiman, 2001).

**Neural Study**  To study the neural representations of phoneme sequences, we conducted single-unit ablation studies by zeroing the output values of units in the recurrent layer (Lakretz et al., 2019; Lakretz, Hupkes, et al., 2021). We then evaluated the performance of the ablated model on the Word Feature Evaluation and Sonority Evaluation Datasets, and compared their results with those of the intact model. To study distributed neural representations across all units, we trained Metric Learning Encoding Models (MLEMs; Jalouzot et al., 2024; Salle et al., 2024), which reveal which linguistic factors best predict neural distances among words.

## Results

### Behavioral Study: Speech Errors and Main Effects in the Neural Model for Word Repetition

**The NWR Model Fully Accomplishes the Word Repetition Task**  LSTMs, but not RNNs, could learn to perform the task

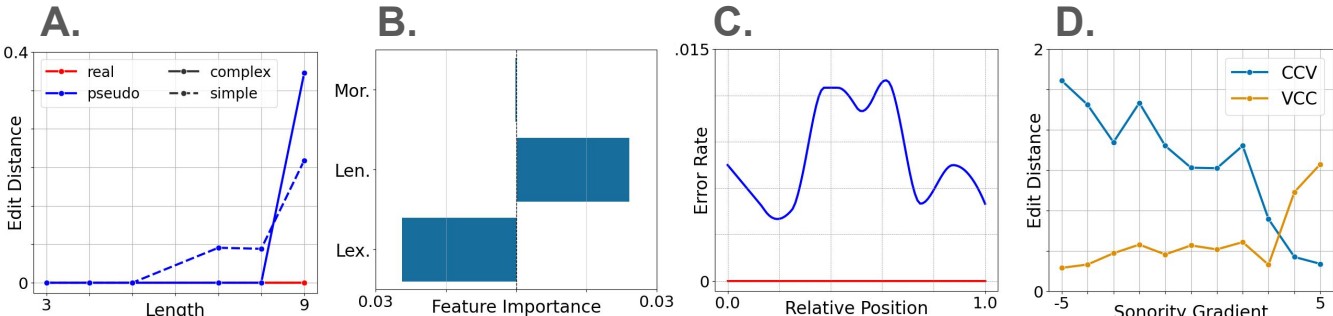

Figure 2: **Speech Errors of the NWR-Model.** We probed model's behavior for several processing effects, known from humans: (A) Length effect and its interaction with lexicality and morphological-complexity. Length effect is observed for pseudowords only. For real words, no errors are expected since the intact (non-ablated) model performs perfect word repetition of real words. An interaction with morphological complexity is observed: Except for nine-phoneme sequences, morphologically-complex words are processed more robustly. (B) Feature Importance (FI) for all main dimensions: lexicality, morphological complexity and word length (frequency was omitted due to its strong correlation with lexicality, assuming pseudowords have zero frequency), which were estimated from a regression model trained to predict edit-distance errors for all words in the Word Feature Evaluation Dataset. The signs of the FIs were determined from the regression coefficients. (C) Error-rate as a function of relative position of the phoneme in the word. A primacy and recency effects are observed: The model tends to make more errors in middle positions compared to early or late ones. (D) Sonority-Sequencing effect: Error-rate as a function of the sonority gradient in a two-consonant cluster, for both CCV and VCC clusters (C - consonant, V - vowel). Overall, the model follows the Sonority Sequencing Principle (SSP), making more errors when sonority gradients violate the SSP.

perfectly on the training data. The hyperparameters of the optimal model among them (see Model Selection) were: *batch size:* $2048$, *hidden size:* $128$, *dropout:* $0$, *learning rate:* $0.001$. This model achieved a zero error rate when trained on the complete lexicon of the Training Dataset. We refer to this selected model as the Neural Word Repetition (NWR) model. Figure 2 shows the performance of this model on our evaluation datasets, which we comment below. To test the robustness of the results, we trained 10 more models from different seeds, using the optimal hyper-parameters from the grid-search. All results are reported in the appendix, showing strong consistency across models.

**Lexicality Effect** The NWR model was able to perfectly reproduce all real words in the test set, which was expected, since all real words appeared in the training dataset. However, the model also perfectly reproduced the vast majority of pseudowords in the Word Feature Evaluation Dataset (97.25%). This suggests good generalization capabilities of the model. This difference between real and pseudowords suggests a lexicality effect (2A; blue vs. red lines), which was significant in the regression model (Figure 2B; $p-value \ll 0.05$; see *Experimental Setup*)

**Length Effect** As seen in Figure 2A, the NWR model makes more errors on longer pseudowords ($\rho = 0.220, p-value \ll 0.05$). This behavior matches what we would expect from a model which employs a mechanism akin to working memory (e.g., a phonological output buffer) for processing words that are not part of an already learned lexicon. The length effect was found significant in the regression model (Figure 2B, $p-value \ll 0.05$).

**Morphological-Complexity Effect** We next asked whether the model made more errors on morphologically simple words, compared to complex ones. This would be expected if morphemes are processed as discrete units, thus reducing the *effective size* of the phoneme sequence. Figure 2A (continuous vs. dashed lines) suggests a morphological-complexity effect: the model started making errors on morphologically simple pseudowords of phoneme-sequence length 7 and greater, and on morphologically complex pseudowords only at length 9. However, the regression model found no significant main effect of morphological complexity ($p-value > 0.05$) or interaction effect with word length ($p-value > 0.05$).

**Primacy and Recency Effect** Next, we studied if the model made more errors at particular positions within the phoneme sequences. Figure 2C shows the error rate distribution for all real (red) and pseudo (blue) words as a function of the position of the phoneme in the sequence (if for a given sequence more than a single error occurred, each error was counted independently). Overall, the model made more errors on phonemes in middle positions compared to positions near the beginning or the end of the sequence. This pattern resembles primacy and recency effects in humans, typical to working-memory processes (e.g., P. Gupta, 2005; P. Gupta et al., 2005).

**Sonority-Gradient Effect** Finally, we studied whether phoneme processing in the NWR model follows the sonority sequencing principle (SSP). Figure 2D shows speech errors made by the NWR model on the Sonority Evaluation Dataset. For CCV syllables, the model made fewer repetition errors on syllables that conform with the SSP (i.e. having a posi-

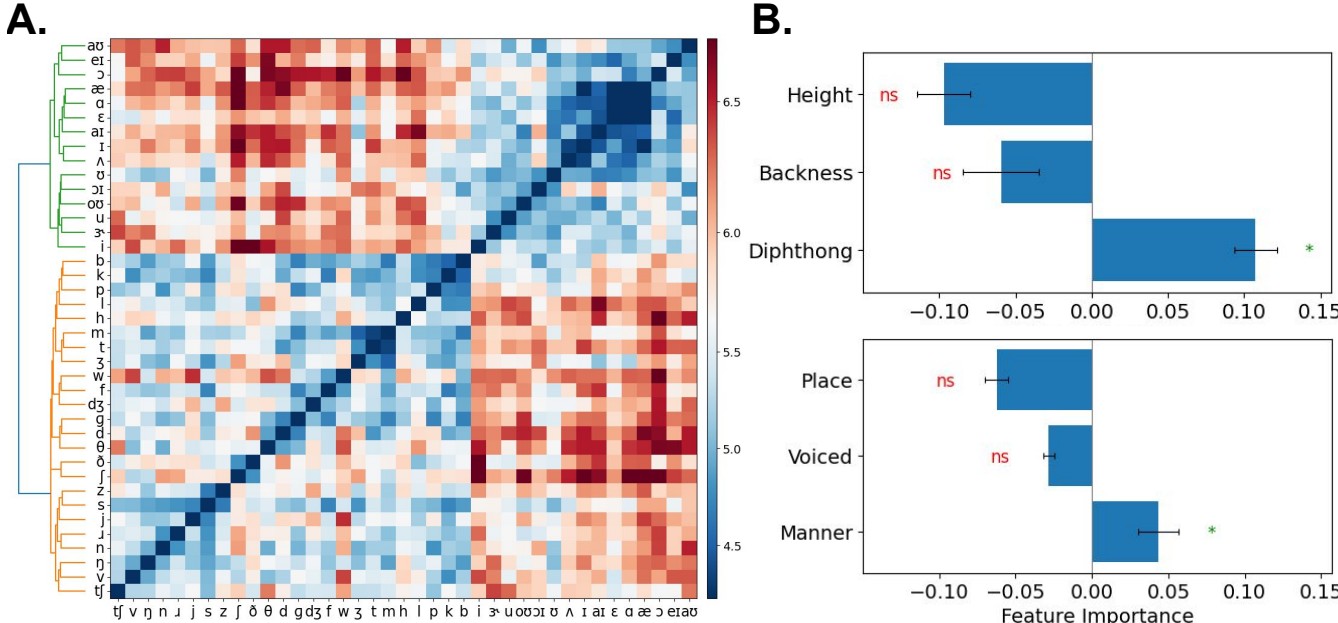

Figure 3: **Neural Representations of Single Phonemes in the NWR Model.** (A) Pairwise Euclidean distances among the 39 phoneme representations, taken from the hidden state of the Encoder after processing each phoneme individually. Rows and columns are sorted based on unsupervised hierarchical clustering (dendrogram on the left). Two macro-clusters are observed, corresponding to vowels and consonants. (B) Feature Importance for vowel and consonant features obtained from a Metric-Learning Encoding Model (Jalouzot et al., 2024). For vowels (top), Height refers the the height of the tongue when pronouncing the phoneme (e.g., high, low). Backness refers to the horizontal placement of the tongue (e.g., back, front). Whether the vowels was a diphthong or not was encoded as a binary variable. Asterisks denote statistical significance; n.s. - not significant. For consonants (bottom), place of articulation is where along the vocal tract the consonant is pronounced (e.g., coronal, labial). Manner of articulation describes the interactions of speech organs to produce a sound (e.g., fricative, nasal). Voiced is a binary feature which encodes whether vibration of the vocal cords is necessary for pronunciation. Error bars are standard error.

tive sonority gradient; $\rho = -0.262, p-value \ll 0.05$). For VCC syllables, the SSP is reversed and so are the results: the NWR model makes more errors with positive sonority gradients ($\rho = 0.114, p-value \ll 0.05$), which is when the SSP is violated.

### Neural Study: Linking Linguistic Features to Neural Representations in the NWR Model

The behavioral effects described above must stem from the model's underlying neural representations and mechanisms for processing phoneme sequences. In this section, we take initial steps toward understanding these by studying single-phoneme representations and conducting ablation experiments on individual model units.

**The Neural Organization of Single-Phoneme Representations in the NWR Model** We first investigated how the NWR model internally represents single phonemes, the basic units of spoken words. Prior research in cognitive science and neuroscience has shown that human phoneme representations are structurally organized. Specifically, during speech comprehension, they are grouped by linguistic features such as manner-of-articulation (e.g., [plosive], [fricative]; Chomsky & Halle (1968)). What kind of neural representations for

phonemes has the NWR developed during training?

To study this, we presented individual phoneme tokens to the Encoder, extracting their corresponding embeddings from its hidden layer. To analyze the pairwise relationships among all phonemes, we computed the Euclidean distances between all embedding pairs. Figure 3A displays the resulting dissimilarity matrix for all phonemes.[5]. This matrix is sorted according to a dendrogram (left side) generated using unsupervised hierarchical clustering (Pedregosa et al., 2011). Remarkably, despite the model receiving no explicit acoustic information during training, it learned to segregate vowels and consonants into distinct regions within its neural space. This clear separation is evident in the two prominent clusters for vowels and consonants visible in the dissimilarity matrix, and also after dimensionality reduction (Figure 13).

Beyond the broad consonant and vowel distinctions, we observed more granular, structured relationships within these groups. For instance, within the consonant clusters, specific sounds such as the plosives /p/, /b/ and /k/ are grouped together. Similarly, among vowels, some diphthongs show clear clustering. However, a straightforward organization based purely on surface phonological features was not immediately

---

[5]Using cosine distance yielded qualitatively similar results.

apparent from the dissimilarity matrices alone.

To determine whether specific phonological features underlie the neural organization of phonemes in the NWR model, we employed Metric Learning Encoding Models. MLEMs are designed to model neural distances from differences in theoretical features, which, in our case, were phonological features. This method assigns a Feature-Importance (FI) score to each phonological feature, quantifying how strongly a difference in that feature predicts a large neural distance.

Figures 3B illustrate the resulting FIs for vowels (top) and consonants (bottom). For vowels, three features were included in the analysis – Height, Backness[6] and whether the sound was a diphthong, which, for simplicity, were encoded in the NWR as single token. MLEMs showed that diphthongs predict the largest neural distances in the Encoder.

For consonants, we contrasted the effects of place, manner-of-articulation, and voicing on neural distances. MLEMs revealed that changes in manner-of-articulation features corresponded to the largest distances in the neural space. This finding aligns with human behavioral observations, where manner-of-articulation features exhibit the greatest discriminative power in English, and also dominate neural representations in the human auditory cortex (Mesgarani et al., 2014; Lakretz et al., 2018; Lakretz, Ossmy, et al., 2021).

**Speech Errors of Neurally-Damaged NWR Models** Neuropsychological research has shown that humans can exhibit highly characteristic speech errors after localized brain damage, which can be explained by *selective* impairments to specific components in the cognitive model for word repetition (Figure 1A-top). If during training, the NWR model developed neural circuits akin to the cognitive model, we would expect characteristic speech errors in neurally-damaged NWR models that resemble those reported in humans. Here, we make first steps to test this hypothesis by conducting ablation studies, for which we removed at each time a single unit from the recurrent layer of the NWR model and studied the resulting speech errors in the ablated NWR model.

**Dual-Route Processing in the NWR model?** The single-unit ablation study resulted in 128 ablated NWR models. Figure 4 summarizes errors made by all 128 ablated models on the Word Feature Evaluation Dataset. Values on the x and y-axes show the percentage of errors made by the ablated model on real and pseudo words, respectively. Each dot represents a different ablated NWR model.[7]

If single-unit ablation were to lead to some ablated models appearing in the lower triangular region of the plot (i.e., making more lexical errors), while other models appeared in the upper triangular region (i.e., making more sublexical errors), this would provide support for the emergence of dual-route processing within the NWR model. Of course, the absence

---

[6]Due to a strong correlation between Backness and the Roundedness feature, the latter was omitted from the regression model.

[7]Complete results for all other features pairs (e.g. short/long) can be found in Figure 7 in the Appendix.

of such evidence is not evidence of the absence of dual-route processing, but a positive result would offer strong support. What did we find in the NWR model?

Figure 4 shows that most ablated models had an error rate under 20% on the evaluation dataset, showing general robustness to ablation. However, single-unit ablations resulted in higher error rates when performed in the Encoder (blue) than in the Decoder (red). This suggests that sequence encoding uses smaller, less redundant circuits than sequence production.

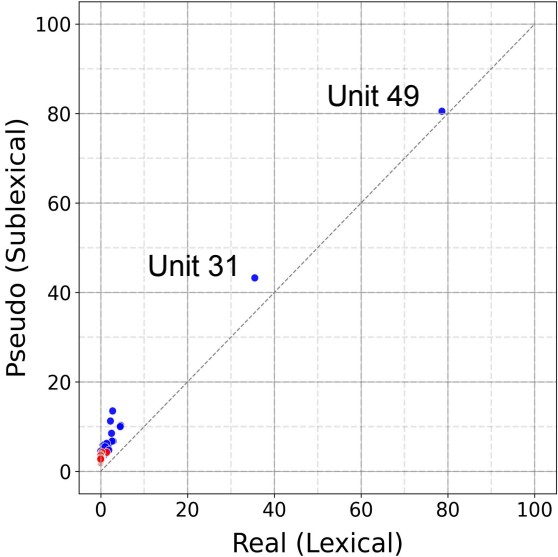

Figure 4: **Speech Errors of all ablated models.** We performed a single-unit ablation study where each hidden layer unit (blue for Encoder, red for Decoder) was ablated one at a time. The model's performance was then re-evaluated on the factorial dataset. Each dot represents a single ablated model. The axis values indicate the percentage of error following ablation, specifically contrasting real vs. pseudowords from the factorial dataset. Units on the diagonal had a similar effect on real and pseudowords; units in the upper triangle caused more sublexical errors.

Figure 4 further shows that all ablated models lie close to the diagonal, in the upper triangle of the scatter (see Appendix for the distribution of the distances from the diagonal). That is, single-unit ablations cause more errors in sublexical rather than in lexical processing. This suggests only a single, rather than double, dissociation between the two routes of the cognitive model.

Interestingly, two units (number 31 and 49) caused a large increase in speech errors, with one unit (49) causing error rates up to 80% for both real and pseudo words, as discussed next.

**Speech Errors Following the Ablation of Unit 49** Given the large effect on error rate following the ablation of unit 49, we conducted an in-depth analysis of the corresponding

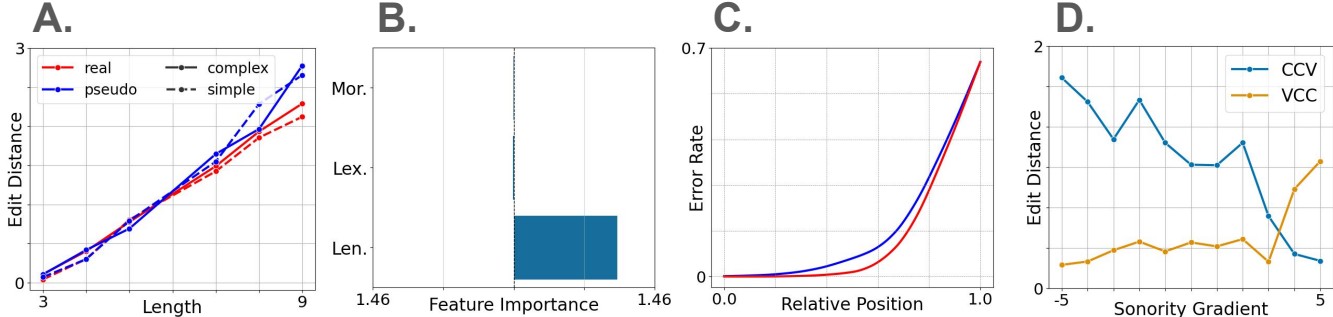

Figure 5: **Speech errors of the NWR model following the ablation of unit 49.** Panels are organized as in Figure 2

ablated NWR model. Fig. 5 shows the behavioral analysis of the ablated model, with its performance in the different conditions.[8] We highlight several key observations: In Figure 5A, we find a strong length effect ($\rho = 0.665, p-value \ll 10^{-3}$), but no lexicality effect (with similar difficulty for real and pseudo words) and no morphological-complexity effect. Figure 5B further quantifies this, showing that the length effect dominates the results. Furthermore, Figure 5C shows that ablating unit 49 eliminates the recency effect of the original model, and 5D shows that the sonority-gradient effect is preserved for both CCV and VCC syllables. An analysis of the error patterns of this unit revealed that the model prematurely stops sequence producing during word repetition (see Appendix D). This premature stopping of sequence production explains the strong length effect (panels A&B) and the absence of a recency effect (panel C).

## Discussion

We introduce a novel approach to investigate whether the dual-route cognitive model for word repetition can be mapped onto neural processing within an artificial neural model trained on this task. We propose three new evaluation methods: (1) assessing human-like linguistic behavior in the neural model using a defined set of criteria; (2) determining the emergence of dual-route processing by contrasting lexical and sublexical errors; and (3) performing component-wise tests of the cognitive model's individual parts, by providing a list of expected effects for each component.

Unlike previous studies, we trained our models on a large lexicon including polysyllabic and multi-morphemic words, also accounting for the natural frequency of words. Our results show that the neural word repetition (NWR) model successfully learns to reproduce all words in the training lexicon and can accurately repeat most pseudowords in the evaluation test. The model exhibited several human-like processing effects, including a length effect, primacy and recency effects, and adherence to the sonority sequencing principle. However, it did not show a sensitivity to morphological complexity.

Our analysis of the NWR model's single-phoneme repre-

sentations revealed an organization into distinct vowel and consonant clusters. This structure emerged during training based solely on phoneme co-occurrence statistics, without any acoustic input. We found that manner-of-articulation features most strongly predicted neural distances for consonants, while the diphthong vs. monophthong distinction was key for vowels. In the above respects at least, the representations of phonemes by the NWR model were consistent with human processing.

To investigate whether dual-route processing emerges during training, we conducted ablation studies, simulating neural damage in the model. The resulting 'patient' models displayed a tendency to make both lexical and sublexical errors. As seen in Figure 4, most ablated models clustered around the diagonal, suggesting that ablating a single unit tended to impact both lexical and sublexical processing similarly, or at least caused more errors in sublexical processing, but not the other way around. This pattern indicates a single dissociation between lexical and sublexical processing, or potentially even no clear dissociation, at least not at the single-unit level. Consistently, a follow-up analysis of how lexical and sublexical information is encoded by different units of the model did not reveal distinct, lexicality-based separation of units (Figures 12&14). These results more closely align with the view that lexical and sublexical processing are entangled, exhibiting no sharp boundaries between them (e.g., Regev et al., 2024).

Overall, this study takes initial steps toward bridging the gap between the cognitive model of word repetition and its underlying neural mechanisms in the human brain by developing a neural model that can be analyzed at both behavioral and neural levels. By training the model on a large lexicon and systematically examining its behavior, we provide evidence that key human-like processing effects can emerge during training, including those related to working memory, without explicitly introducing working-memory dynamics into the model. Future research should investigate how lexical and sublexical information is represented across different units of the model, how phoneme sequences are neurally encoded and whether dual-route processing can be more explicitly induced by incorporating working-memory dynamics into the model, or by adding architectural constraints inspired by human neuroanatomy.

---

[8]see Appendix for the speech errors of the NWR model following the ablation of unit 39.

## Acknowledgments

This work was supported by grants ComCogMean (Projet-ANR-23-CE28-0016), FrontCog (ANR-17-EURE-0017), and ANR-10-IDEX-0001-02. It was performed using HPC resources from GENCI–IDRIS (Grant 2024-AD011015802). Special thanks to Ali Al-Azem and Louis Jalouzot for their valuable feedback.

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

## Appendices

**A  Word Feature Evaluation Dataset**  The length of each word is the number of phonemes, and not the number of letters, it contains. Short words have between 3, 4, and 5 phonemes. Long words have between 7, 8, and 9. Low frequency (real) words have a Zipf frequency up to 3.5, and high frequency words have Zipf frequency as low as 4.0. A word is morphologically complex if it contains a prefix or suffix appended to a distinguishable root (e.g. restart is complex, but repeat is simple, because -peat does not stand on its own in the same sense).

**B  Model Selection**  Before settling on the hyperparameter ranges for our final grid search, we first ran a series of preliminary tests to explore the hyperparameter space. To facilitate training by preventing the accumulation of prediction errors, we implemented a teacher-forcing procedure. That is, the predicted token is replaced with the ground-truth token, with a given probability. We ultimately found its effects to be detrimental to learning. Far more significant are the effects of the learning and dropout rates. We found no clear advantage in increasing the number of recurrent layers beyond one layer.

We explored all combinations obtained for variations of hidden size (64 or 128), dropout rate (0, 0.1, 0.2), batch size (1024, 2048, 4096) and learning rate ($5.10^{-3}$, $10^{-3}$, $5.10^{-4}$). We followed a 5-fold cross-validation (CV) procedure. Each CV split contained 30k training words sampled by frequency to generate $10^6$, as in the complete training data set. To avoid overfitting, we used early stopping, choosing the $75^{th}$ epoch (the middle of the period between epochs 65 and 85, when the model first achieved a stable zero error rate on the Training Dataset)

| Hyperparameter | Range |
|---|---|
| # of Layers | $1 - 2$ |
| Hidden Size | $2 - 512$ |
| Dropout Rate | $0.0 - 0.7$ |
| Learning Rate | $10^{-5} - 10^{-2}$ |
| Teacher-Forcing | $0.0 - 0.7$ |

Table 2: Hyperparameter ranges for first grid search.

We settled on two hidden sizes for the grid search, 64 and 128. This equates to 256 (512) hidden units for both the encoder and the decoder, so 512 (1024) hidden units total for RNNs and LSTMs, respectively. We found a handful of promising models with hidden size 64 that had not yet converged at the end of 100 epochs. Many of them learned to complete the task perfectly on the training set after a second round of training for 150 epochs. The results of the model among them with the earliest stable zero error rate are included below. Ablation on this model revealed another neuron whose ablation caused the same behavioral effect as the ablation of neuron 49 in our chosen model of hidden size 128 discussed above. That is, a consistent premature emission of End-of-Sequence tokens.

## C  Analysis of the NWR model

**Types of error**  The edit distance is computed on the basis of 3 operations : insertions, deletions, and substitutions. We kept track of the average number of each operation per word for every condition possible in our Word Feature Evaluation Dataset. Those numbers are reported on Figure 8A.

## D  Ablation Study

**Ablation of Unit 49**  Given the results reported on Figure 5, we examined the predicted phonemes over the factorial design dataset. We observed that this ablated model was most often outputting end of sequence tokens (`<EOS>`) from position 4 to 8 and up to the end of the word with seemingly no correlation with other factors than length. Some phoneme substitutions could also be observed sporadically, although no pattern was easily identifiable. For example:

`[F, R, EY, M, W, ER, K] → [F, R, EY, M, W, ER, <EOS>]`

`[F, IH, SH, IH, NG] → [F, IH, SH, <EOS>, <EOS>]`

**Ablation of other units**  Fig. 15 reports a categorization of errors induced by ablating the most significant neurons. Only neurons inducing at least 50 errors are reported.

## E  Replication across Model Seeds

To test the robustness of the results, we trained 10 more models with the same architecture, same hyperparameters, with 10 different seeds. First, we found that all versions of the model reached zero errors on real words in the train dataset for the first time as early as epoch 39 and as late as epoch 85. Figure 10 shows the average results across models, with shaded areas as the 95% confidence interval across model seeds, demonstrating the robustness of all identified effects across model seeds.

We also repeated the ablation study on the 10 new replicate models. In every case, we found that only 2 to 4 units had a significant impact ($> \%20$) on model performance upon ablation, this almost always being units in the encoder.

The unit with the strongest effect showed a typical behavior across all seeds, similar to that of Unit 49 in the original NWR model. The ablation of this unit, in all models, caused a typical length effect (c.f. Figure 11). Moreover, closer inspection of ablated models' predictions showed that these errors were of the same kind: premature prediction of the end of the word.

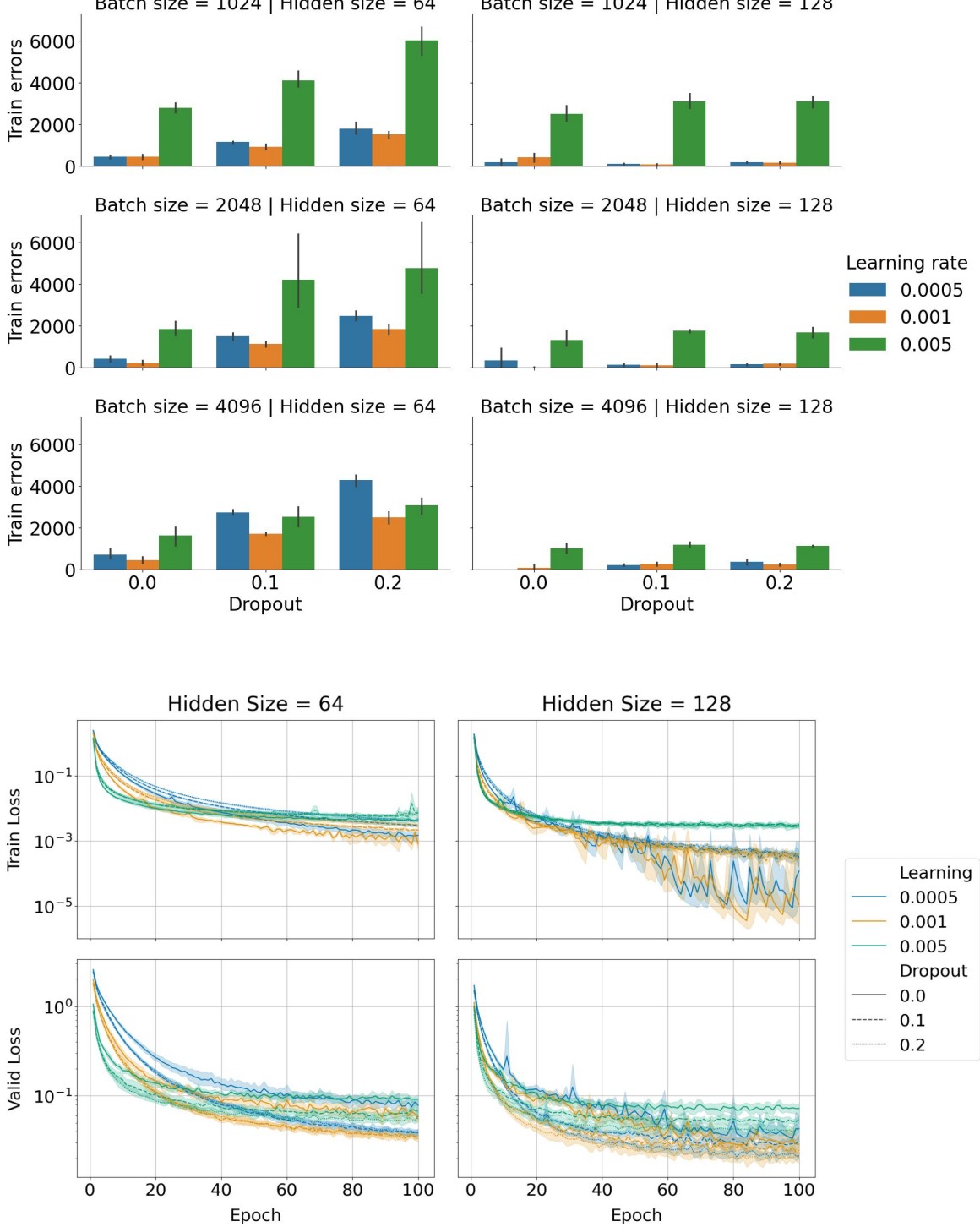

Figure 6: **Grid search results.**

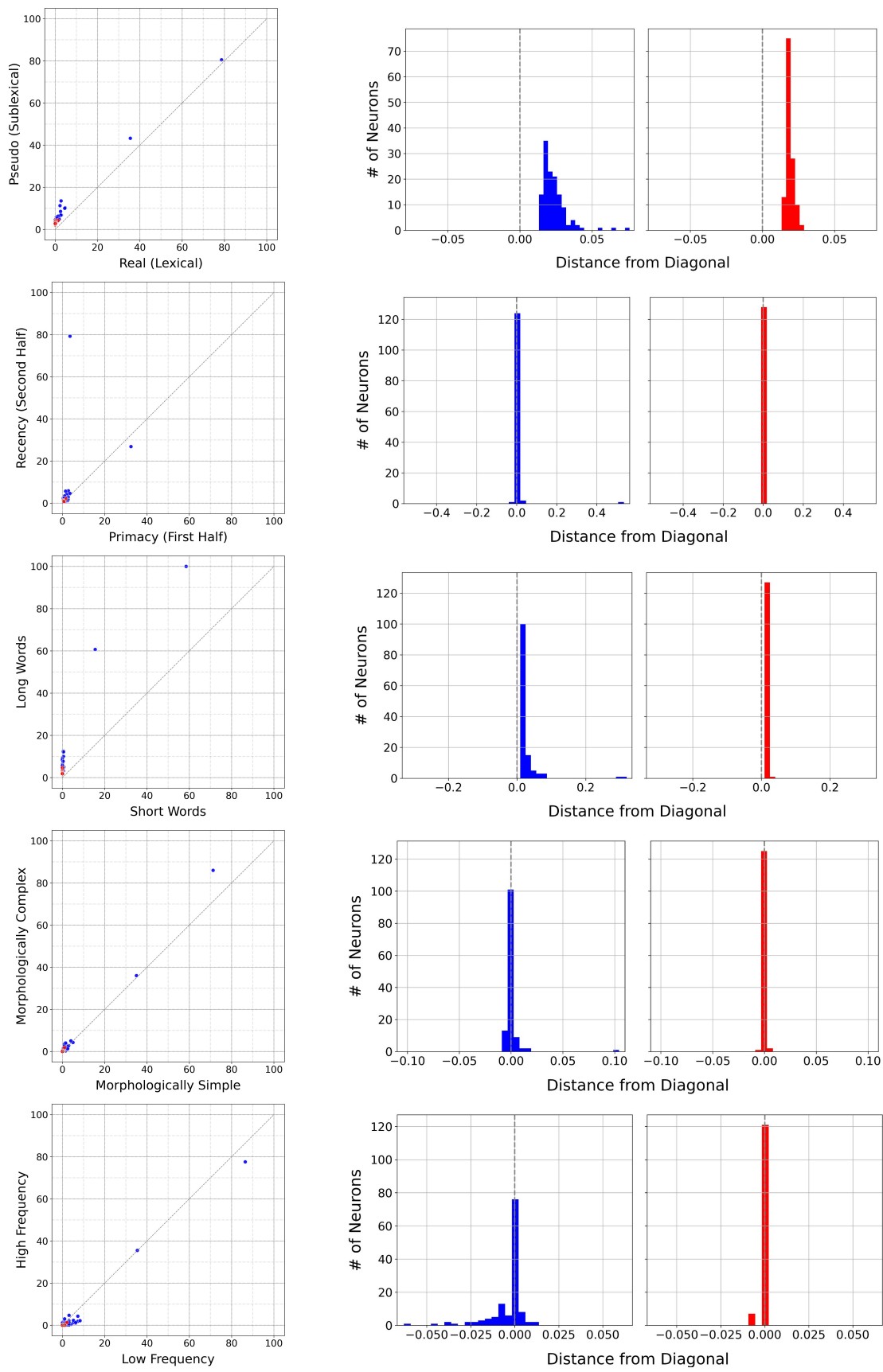

Figure 7: **Full results for ablation study on NWR model, organized by factor.**

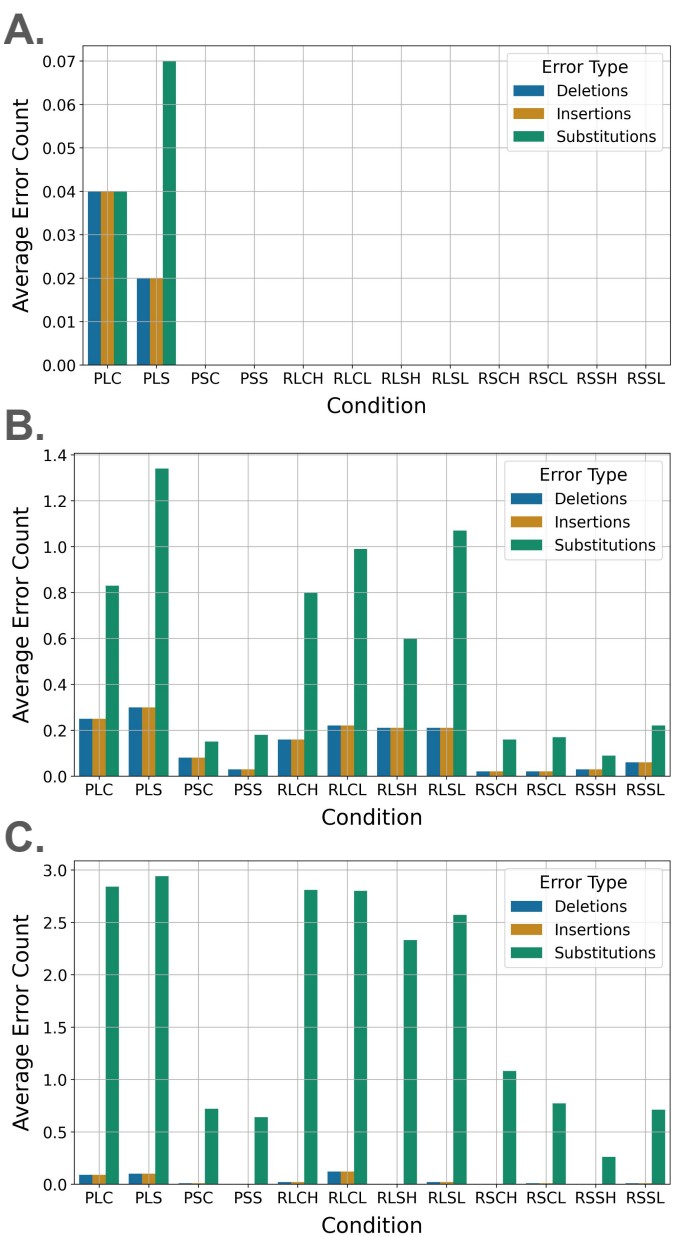

Figure 8: **Types of errors of the NWR model** Type and average count of errors made by the NWR model depending on word condition. Condition is made of the initials of lexicality (**R**eal, **P**seudo), length (**L**ong, **S**hort), morphology (**C**omplex, **S**imple) and when relevent, frequence (**H**igh, **L**ow). The types are determined from the Levenshtein distance computation. (A) Error types of the NWR model. (B) Error types for NWR model with unit 49 ablated. (C) Error types for NWR model with unit 31 ablated.

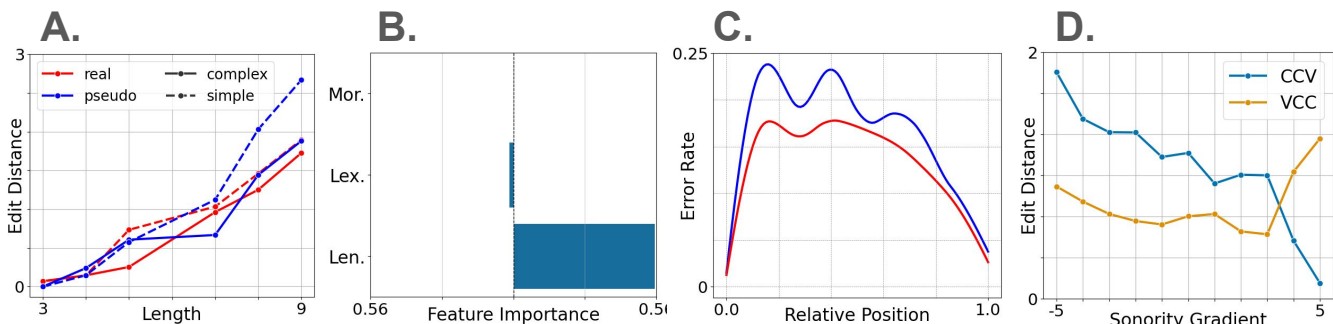

Figure 9: **Speech errors of the NWR model following the ablation of unit 31.** Same as Figure 2

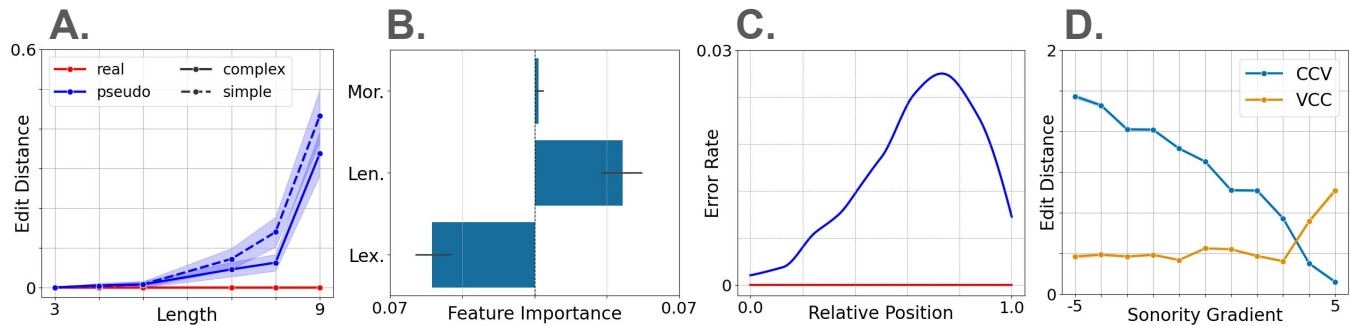

Figure 10: **Mean speech errors over 10 seeds.** Same as Figure 2. Error bars reflect standard error.

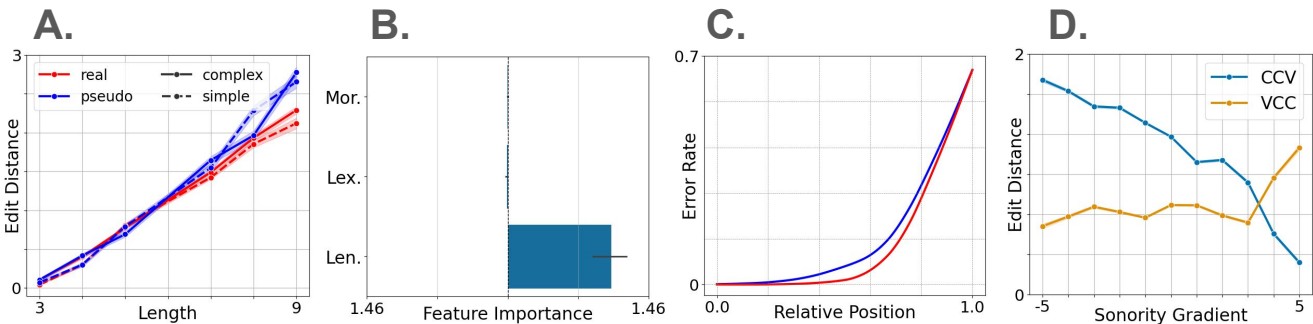

Figure 11: **Mean speech errors of 49-like units over 10 seeds.** Same as Figure 2. Error bars reflect standard error.

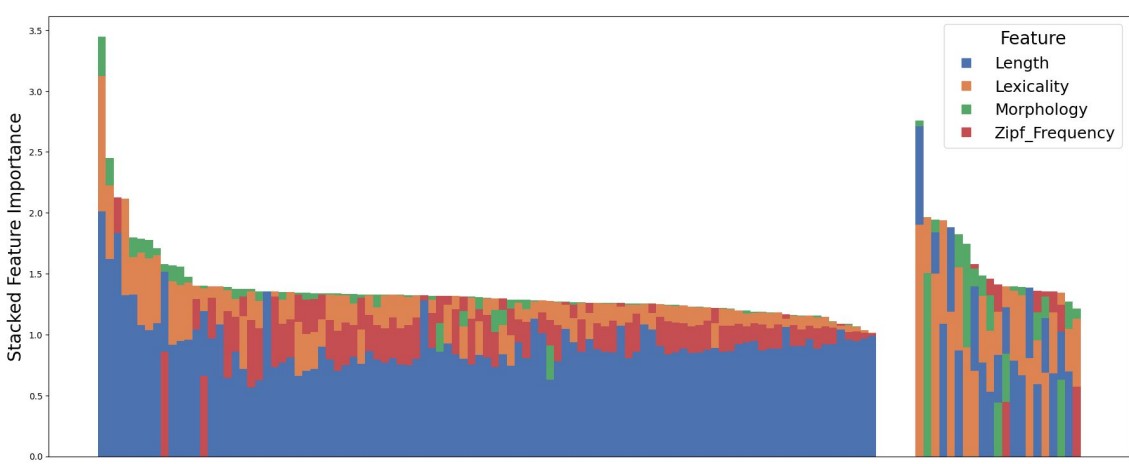

Figure 12: **Stacked feature importance per neuron** Neurons were clustered according to their feature importance profiles using k-means. The optimal number of clusters was computed by comparing the silhouette scores of the results of k-means clustering the neurons with k = [2,8]. The first group identified is clearly dominated by length, which is consistent with the regression analyses. The profile for the second cluster is more difficult to interpret. Yet, overall, an increase in the importance of lexicality relative to the other features is observed.

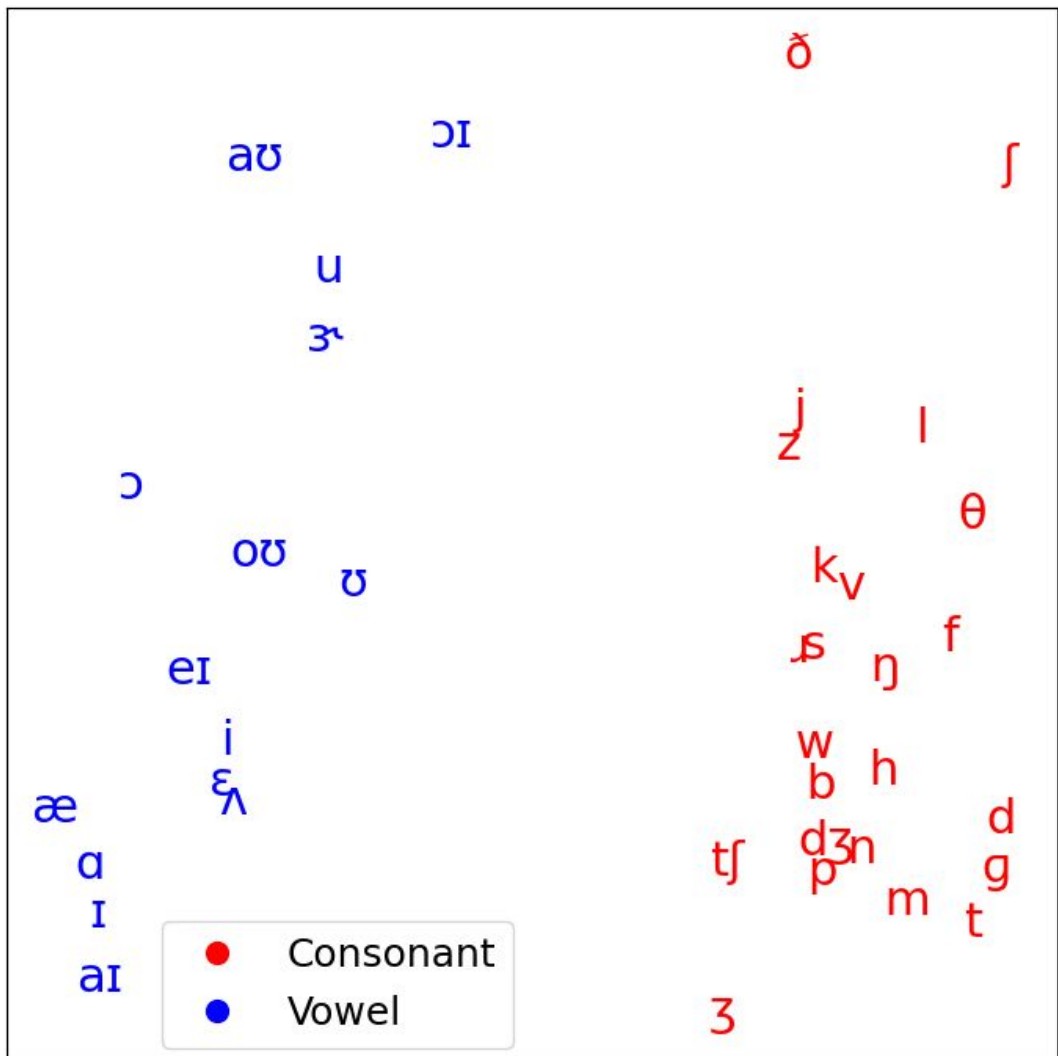

Figure 13: **Dimensionality Reduction of the Pairwise Distances among all Single-Phoneme Representations**

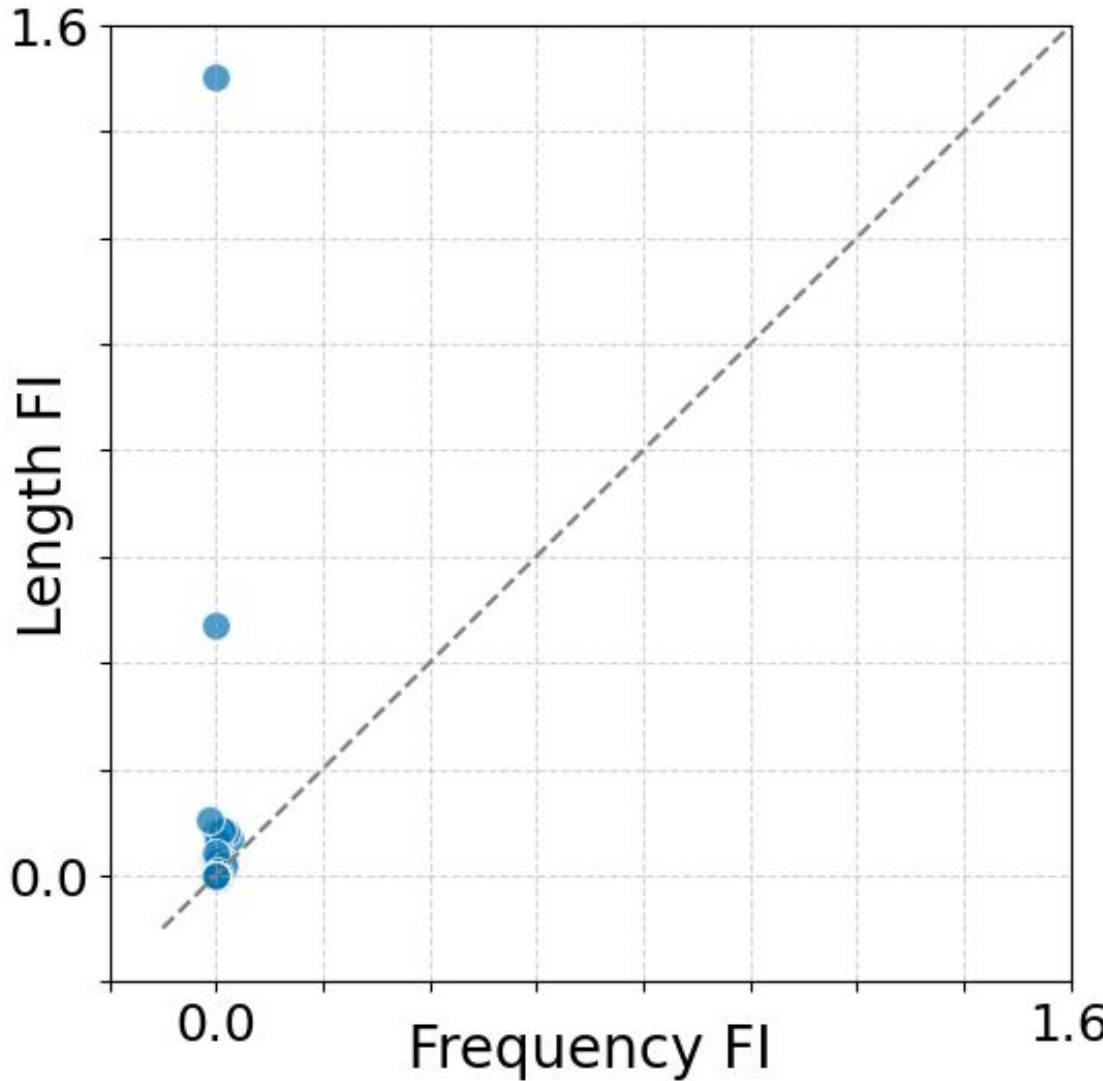

Figure 14: **Feature Importances for Length and Zipf Frequency across all ablations of NWR model** This figure shows the feature importances for length and frequency for modeling the errors of each ablated NWR model. The significant FIs for length correspond to units 49 and 31. We found no model where the FI for frequency was significant.

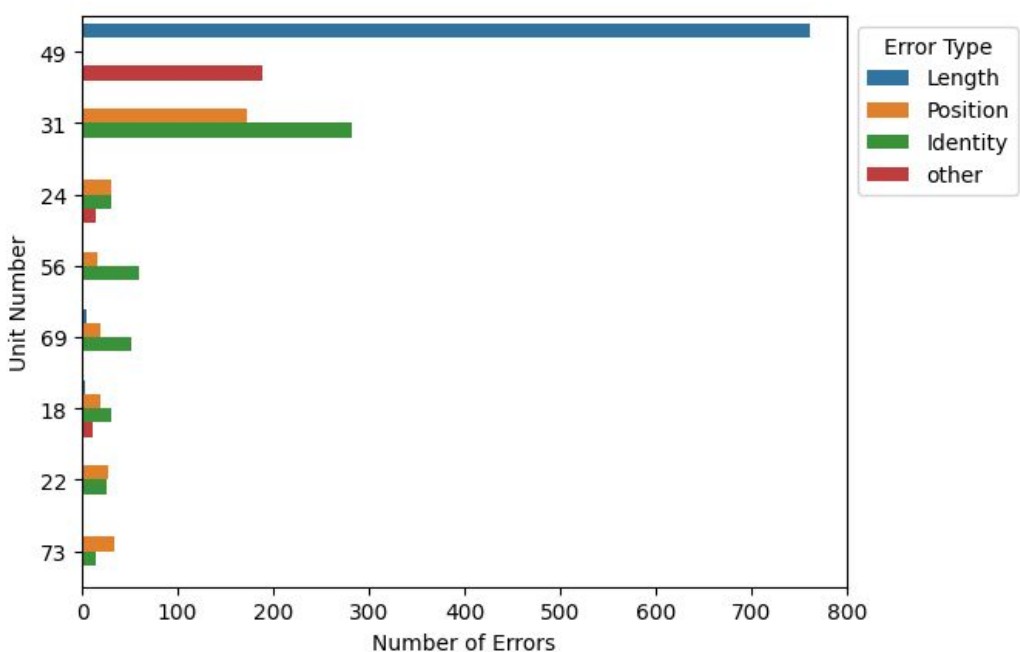

Figure 15: **Classification of Errors for Ablated units in the NWR Model** All single-unit ablations resulting in at least 50 errors are included in this figure. Length error : premature prediction of `<EOS>` token with all previous phonemes being correct. Position error : error resulting from the position of phonemes being confused by the model, while preserving all phoneme identities (i.e. the prediction is a permutation of the initial sequence). Identity error : at least one phoneme being substituted with another, in a given position. Other : any error not falling in the previous categories. We observe that ablation 49 cause length errors. Ablation 31 has a combination of position and identity errors. These position errors are generally the instances where vowels were permuted.

