# OpenReview forum: "A Neural Model for Word Repetition"
_ccneuro.org/CCN/2025/Proceedings — CCN 2025 Proceedings asProceedingsPoster_

### Official Review · Reviewer_xbCQ · 2025-03-26
**Review for #115**

**Soundness:** 2
**Clarity:** 3

**Comments:**

This paper investigates word repetition using a neural model inspired by cognitive and neuropsychological theories. The authors implement an encoder-decoder RNN/LSTM architecture to simulate word repetition and examine whether key human-like effects (e.g., lexicality, frequency, and working memory constraints) emerge in the model. They further conduct ablation studies to mimic brain damage, analyzing how removing specific neurons impacts model performance. Their results suggest that the model captures some cognitive effects observed in humans, such as primacy/recency effects and the sonority sequencing principle, but diverges in other areas, such as morphological complexity processing.

Strengths:
- Overall writing is great. The methodology is well-documented, with training datasets, test conditions, and model details clearly outlined.
- The model is evaluated using carefully controlled test datasets that reflect known psycholinguistic effects, allowing for a rigorous behavioral assessment.
- The neural lesioning experiments provide interesting insights into how different units of the model contribute to speech processing, offering potential relevance to aphasia research.

Concerns:
- The model doesn’t functionally map to different brain regions or pathways, despite documenting in the introduction how the brain deals with similar tasks. This makes it difficult to justify and assess the usefulness of the model for understanding the neural mechanisms. With this said, the model still produces interesting effects that are observed in humans.
- The authors spent a significant chunk of the beginning of the paper on discussing dual-route processing, but did not successfully map the concepts into their models, which makes reading slightly confusing. As discussed in the limitation section, the model doesn’t seem to address any of the LTM properties. Changes of how to frame the motivations of the work might be needed.

**Expertise:**

1

**Interest:**

2

---

> ### Author Rebuttal · Authors · 2025-04-15
>
> We would like to thank the reviewer for the thoughtful and useful feedback and the acknowledgment of the strengths of our paper.
>
> ### Mapping to different brain regions or pathways and discussion of dual-route processing in the paper.
> Please see our response to the other reviewers for further details. Specifically, this work is the initial investigation into how well deep neural models simulate human word repetition. Consequently, we tested the most parsimonious hypothesis: that dual-route processing arises naturally within a model without pre-defined structure during training, instead of assuming inherent dual routes. Mapping to different brain regions was therefore not desired at this stage. Our paper offers a thorough evaluation of this hypothesis. Observing that the ablation of one neuron increased real word errors while another increased pseudoword errors would have indicated a double dissociation between the two processing pathways. Taken together, this work motivates future steps, to evaluate what are the necessary augmentation of the model to better capture dual-route processing as observed in humans, while preserving parsimoniousness.
>
> We now make the discussion of this point, and the overall hypothesis testing in the work, clearer in the paper.

---

> > ### Comment · Reviewer_xbCQ · 2025-04-21
> >
> > Thank you to the authors for the thorough and thoughtful rebuttal. I appreciate the additional analyses and clarifications provided, especially regarding the consistency of ablation results across seeds and the reasoning behind the lexicality analyses.
> >
> > Given that my background is not specialized in this domain, I don’t feel well-positioned to confidently judge the broader theoretical implications. From my perspective, the authors have adequately addressed my earlier concerns, and I’m happy to defer to the judgment of the other reviewers who are more expert in this area.

---

### Official Review · Reviewer_uopa · 2025-03-28
**The manuscript "A Neural Model for Word Repetition" is a well-written study investigating if a well-recognized machine learning model can be analogous to the neural implementation of a cognitive dual-route model. This application of an AI system as a theory, not as a tool, was capable of producing the expected patterns shown in lesion studies. Unfortunately, I have severe reservations about the interpretation of the results. The manuscript has several significant issues, which are described below.**

**Soundness:** 1
**Clarity:** 2

**Comments:**

The manuscript "A Neural Model for Word Repetition" is a well-written study investigating if a well-recognized machine learning model can be analogous to the neural implementation of a cognitive dual-route model. This application of an AI system as a theory, not as a tool, was capable of producing the expected patterns shown in lesion studies. Unfortunately, I have severe reservations about the interpretation of the results. The manuscript has several significant issues, which are described below.

Model selection and interpretation: Figure 1 aligns a cognitive dual route structure with a complex encoder/decoder machine learning structure directly interpreting the LSTM architecture to represent different cognitive modules. Using a machine learning model as a theoretical model is controversial (e.g., see Wichmann, F. A., & Geirhos, R. (2023). Are deep neural networks adequate behavioral models of human visual perception?. Annual review of vision science, 9(1), 501-524.), and the current study here misses a key necessity: Model comparison. For example, the obvious comparison would be with models that implement the two routes separately. Thus, one can conclude that the architecture used here is essential for the model results. In the end, based on some findings (i.e., in the  Ablation results), the authors ended up with the interpretation (i.e., discussion in l512) that the removal of single units resulted in errors that typically result from either one or the other route.


l170-184: Provide literature for benchmark effects 1-3

l256: It is unclear why one wants to diminish the variance components to a factorial design. The best would be to use a parametric design here, as a regression analysis is also performed later.

l263-280: This selection of pseudowords makes the task harder for the pseudowords; thus, simply the stimulus selection would allow predicting higher error rates for the selected pseudowords. More interesting would be to use pseudowords with the same characteristics (i.e., letter strings with a highly similar Levenshtein distance, i.e., overlapping distributions). This difference could also be a significant issue for the study that is tried to be prevented in experimental design in humans (i.e., matching of word/non-word characteristics, see Vinckier, F., Dehaene, S., Jobert, A., Dubus, J. P., Sigman, M., & Cohen, L. (2007). Hierarchical coding of letter strings in the ventral stream: dissecting the inner organization of the visual word-form system. Neuron, 55(1), 143-156.)

l363: Missing reference that shows that parameter correlations less influence Feature importance. Again, using a parametric frequency version would allow a better estimation.

Results in Fig. 2: The perfect performance of the actual words concerns me, especially in combination with the designed differences between the two sets of stimuli, potentially leading the model to overfit and thus be more error-prone to the, by design, different stimulus conditions.

Ablation analysis: This part is hard to follow, as a clear description of the analysis procedure is missing. Also, the scaling of Fig.3 is odd. It looks like the analysis also significantly changes the performance of words.

Selective reporting. Please provide all the effects for both dependent measures, similar to those in Fig. 2 A/C.

**Expertise:**

2

**Interest:**

1

---

> ### Author Rebuttal · Authors · 2025-04-15
>
> ### Are deep neural networks adequate models of human behavior?
> Our paper is the first in a line of work to evaluate the extent to which deep neural models can capture human-like word repetition. It therefore tests the simplest hypothesis, lightest in terms of assumptions, whereby dual route processing spontaneously emerges in an otherwise unstructured model during training, rather than assuming hard-coded dual routes. The paper provides a comprehensive evaluation of this hypothesis. Had we found that one neuron increased errors on real words when ablated, and another increased errors on pseudowords, then this would have constituted a double dissociation between the two routes. Furthermore, following R1's comment, we added an analysis with greater sensitivity compared to the ablation approach. Given that our study concluded on the rejection of the main hypothesis, this motivates modelling pathways for future work.
>
> ### Parameterizing the design
> We fully agree that parametrization is desired. In fact, this is precisely the case for our regression analyses. Both word length and frequency were already parameterized. We now better clarify this point in the manuscript. Parameterizing other factors, such as lexicality and morphology is indeed interesting. We opted to keep them binary in this first study, for the sake of simplicity, but we will explore further in future work.
>
> ### Perfect performance on real words is desired
> To model human behavior, the model, by design, needs to succeed on lexical words. Pseudowords are therefore inherently more difficult for the model, just like for humans. By having the model memorize the “real” words, we simulate the building of a lexicon, which should be subject to LTM effects. We then expose the model to pseudowords, withheld from training, to assess the WM of the model. Here it cannot fully rely on its memory from training to reconstruct the input. Importantly, the fact that the trained model achieves 97% performance on pseudowords highlights good generalization capabilities and no overfitting. Note that we employed several methods to avoid overfitting, such as drop out, limited model complexity and early stopping.
>
> ### No selective reporting
> Fig. 4 in the main text and Fig. 8 in the appendix are the same plots as Fig. 2, but after ablating neurons 49 and 31 respectively. We take the reviewer’s reference to “both dependent variables” to mean edit (Levenshtein) distance and error rate, which are included in both figures.

---

> > ### Comment · Reviewer_uopa · 2025-04-22
> >
> > Thank you to the authors for their thoughtful rebuttal and for providing the additional figures in the appendix—we appreciate these additions.
> >
> >
> > However, I remain unconvinced by the response regarding the central theoretical concern. The authors describe the model as one that "tests the simplest hypothesis," yet a single-route model would constitute the simpler theoretical alternative. As such, the current model can only support the authors' claims when explicitly contrasted with such a baseline. I expected a more direct engagement with this issue in the revised manuscript, but unfortunately, the discussion does not address it.
> > The interpretation of the LSTM architecture as a dual-route mechanism raises further concerns. As a previous reviewer noted, this analogy may be too speculative and risks being misleading. Given that the current results do not demonstrate an apparent double dissociation, I believe this interpretation should be addressed more cautiously and discussed explicitly, especially since the authors describe this paper as the beginning of a broader research program.
> >
> > Related to this, it would be helpful for the discussion to reflect more on the broader debate about using machine learning models as proxies for cognitive or neural mechanisms. Acknowledging that the model was designed primarily to solve a technical problem rather than to instantiate a formal cognitive theory could help clarify its intended contribution and mitigate some of the theoretical concerns.
> >
> > To clarify a previous point regarding selective reporting, while the error rate is consistently used to analyze relative position effects, the edit distance is not. This discrepancy remains in all relevant figures and would benefit from more apparent justification or correction.
> >
> > In addition, a few relatively straightforward issues raised earlier appear to have been overlooked:
> > - Key references for well-established effects, such as lexicality, frequency, and word length, are still missing.
> > - The reference supporting the claim that parameter correlations have limited influence on feature importance is missing.
> > - Finally, the decision to include only orthographically distant pseudowords from real words warrants further explanation. This choice seems central and should be explicitly justified.
> >
> > I hope these comments are helpful in further improving the manuscript.

---

> > > ### Author Response · Authors · 2025-04-22
> > >
> > > Thank you for your comment.
> > >
> > > The current implementation is indeed looking as well at the single route hypothesis. No simpler implementation (i.e. no simpler neural network) can be used. The study here asks whether or not a dual route emerges for these simple models. These models are constituting a baseline, more specialized model (leveraging dual route architecture) can later be compared to this model. Hence, it remains important to assess whether or not this baseline exhibits dual route processing for future work.
> > >
> > > The study does not claim that LSTM can be interpreted as a dual route mechanism, but explores whether or not this is possible. Indeed, the current study didn’t highlight a double dissociation, requiring more in depth work to completely rule out the existence of that double dissociation.
> > >
> > > Concerning using or not the edit distance for localised errors, this is simply tied to the nature of edit distance. Edit distance relies on alignment and counts operations like insertion, deletions and substitutions necessary to change one string to another. Edit distance is then the minimum number of operations required in those kinds of alignments. It is important to highlight that many alignments might lead to the same number of operations, without having operations happening at the same places, or operations even being of the same nature. Hence, choosing a specific alignment would induce a bias, and averaging them might flatten out biases that the model might have (as well as induce a computational cost). Hence, the choice has been made to not use edit distance for localized studies, as this kind of studies would have been flawed by design.
> > >
> > > We think papers mentioning those effects might have been mentioned in the introduction without being recited when listing the effects. For sources we might include in the final paper, there are :
> > > - https://doi.org/10.1080/14640749208401283 (for word length, lexicality, frequency)
> > > - for the fact that parameter correlations have limited influence on feature importance, it is tied to the fact that parameters might be regularized (e.g. in Lasso) and not represent an actual measure of importance, that it is hard for the model to disentangle effects of coinciding effects in the coefficients, while feature importance is based on predictive power of the model and not the actual values of the data. Permutation importance measures the drop in performance when breaking the correlation between one feature and the others by permuting the value of this feature. However, it remains true that if two features are very highly correlated, they might remain correlated at some level after any permutation.
> > >
> > > For orthographically distant pseudowords, it is important to know that the model has no access to orthography and is only fed phonemes. In that sense, it is only natural that the pseudowords are designed from the phonotactics. However, it is important to say that including pseudowords closer to lexicalized words as well as non-words could be useful to gather broader results in future works.
> > >
> > > Thanks again for your comments that are indeed helpful to improve the manuscript.

---

### Official Review · Reviewer_kfFy · 2025-03-28
**Theoretically-motivated model of word repetition**

**Soundness:** 3
**Clarity:** 3

**Comments:**

The paper examines how an LSTM model can capture human-like behaviors for word repetition. To my knowledge, there is not much work in this domain in more "modern" / stimulus-computable models. The authors ground their work in prior literature claiming two different processing routes during word repetition: a lexical route for words stored in long-term memory and a sublexical route for novel words or pseudowords. I cannot speak to the validity of these claims in neuroscience, but the authors' paper is very well-motivated within this framework.

The authors set up a comprehensive, human-inspired evaluation framework which is made up of phenomena from human studies: lexicality effect, frequency effect, length effect, primary/recency effects, and sonority effect. Broadly, the model replicates these phenomena. The authors perform lesioning on the LSTM model's units, and find that lesioning most units do not cause big drops in performance (around ~20%), but two units, and one unit in particular, leads to large drops in performance. I think the fact that lesioning most single units do not lead to massive drops in performance makes sense (robustness), and in some ways the single unit seems like it could be due to some artifact? Also, the authors write "The ablation study did not reveal any neurons which had selectivity for real words." (line 541), but the authors do not describe their motivation for this sentence. Do they expect to find such neurons due to functional modularity of e.g., language?

Overall, this is a very well-written and clear paper. The motivation is solid, and the experiments are well suited to test their claims. My one suggestion would be to train at least a few more models from a different random seed, given that all the authors' claim rely on a single model instantiation.

EDIT AFTER READING OTHER REVIEWS: I do not see a big disagreement with the other reviewers. I agree with the other reviewers' comments, with the one exception that I will fully accept the premise that we can "..use a machine learning model as a theoretical model" (cf. uopa). As I also raised in my original review, "I cannot speak to the validity of these claims in neuroscience", which appears to be a key theme. However, I think even without neural relevance, I think the behavior of the model is worth studying. I also agree that there are some improvements to be done within controlled experimental conditions (e.g. the pseudowords, as raised by uopa) and interpretation of the lesion results (per reply to my original review). That said, I think the work is original and nicely motivated.

**Expertise:**

2

**Interest:**

2

---

> ### Author Rebuttal · Authors · 2025-04-15
>
> We would like to thank all reviewers for their thoughtful feedback, which helped to strengthen the paper.
>
> ### Training more models from different seeds
> Following your suggestion, we retrained the model with 10 different seeds and repeated the ablation study. We found consistency of the ablation results - all 10 seeds had an encoder neuron that when ablated caused the model to prematurely predict the stop token, resulting in significant drops in performance. Most models also produced a neuron or two that when ablated caused the model to both permute vowels within a word across consonants and permute adjacent vowels and consonants. To further support the selectivity to vowels observed in the errors, we added to the appendix a dissimilarity matrix among all phonemes in the dataset, showing a clear separation between vowels and consonants. Note that this consonant/vowel distinction has emerged in the model solely based on phonological regularities, without any acoustic information. The ablation results from the new models mirror what we found with neurons 49 and 31 in the original model, and suggest that the functionality of certain neurons is consistent across seeds.
>
> ### Dual-route processing
> Our remark on line 541 was in reference to the fact that we did not find any neurons that increased error rates for real or pseudowords. We hypothesized a double dissociation, whereby ablating one neuron would increase errors on real words, and ablating another would increase errors on pseudowords, which would strongly suggest that a dual route emerges naturally in the model. To further investigate whether certain neurons were sensitive to lexicality, we added to the appendix an analysis where we compute the permutation importances of the features of the experimental design (Length, Zipf Frequency, Lexicality, and Morphology) in predicting the activations of each neuron at the final step of encoder. So far, we don’t observe strong double dissociation at the level of single units. This motivates two possible directions for future work. First, dual-route processing is highly distributed and therefore cannot be observed with univariate methods. Second, a rejection of our main hypothesis about the emergence of dual-route processing and therefore implementation of a structural bias to the model, to encourage dual-route processing akin to that observed in neuropsychological studies in humans.

---

### Meta-Review · Area_Chair_1KLg · 2025-05-05

**Ccn Recommendation:** Accept as Proceedings

**Metareview:**

As reflected in the diverse reviews, this manuscript is likely to spark debate, especially among researchers interested in the use of machine learning models to study human cognition. It presents a novel and interesting contribution to the field. For those less convinced by this approach, the paper may seem insufficiently critical in supporting its strong claims. However, the use of an LSTM model to study word repetition and the comprehensive analyses provided are of great interest to the CCN community. Even if not all reviewers agree with the authors’ conclusions, I recommend acceptance, as this paper is likely to stimulate and advance discussion in the field.

**Summary:**

This manuscript proposes an LSTM model to study word repetition. The reviews are mixed: while Reviewer 1 praises the use of an LSTM model for investigating cognitive tasks such as word repetition, Reviewers 2 and 3 are more skeptical about this approach. Specifically, they question whether the model can appropriately be interpreted as a dual-route mechanism and find the authors’ interpretations and conclusions too strong given the current empirical evidence. Reviewer 3 was ultimately convinced by the authors’ rebuttals, but Reviewer 2 remained unconvinced and felt that the discussion was not sufficiently critical of the general applicability of machine learning models for the study of human cognition. Nevertheless, all reviewers agreed that the paper is well-written and that the topic is of high interest.

**Expertise:**

2